# Ensuring fair, safe, and interpretable artificial intelligence-based prediction tools in a real-world oncological setting

Renee George[1,4], Benjamin Ellis[1,4], Andrew West[1], Alex Graff[1], Stephen Weaver[1], Michelle Abramowski[1], Katelin Brown[1], Lauren Kerr[1], Sheng-Chieh Lu [2], Christine Swisher[1,3,5] & Chris Sidey-Gibbons[2,5✉]

## Abstract

**Background** Cancer patients often experience treatment-related symptoms which, if uncontrolled, may require emergency department admission. We developed models identifying breast or genitourinary cancer patients at the risk of attending emergency department (ED) within 30-days and demonstrated the development, validation, and proactive approach to in-production monitoring of an artificial intelligence-based predictive model during a 3-month simulated deployment at a cancer hospital in the United States.

**Methods** We used routinely-collected electronic health record data to develop our predictive models. We evaluated models including a variational autoencoder $k$-nearest neighbors algorithm (VAE-$k$NN) and model behaviors with a sample containing 84,138 observations from 28,369 patients. We assessed the model during a 77-day production period exposure to live data using a proactively monitoring process with predefined metrics.

**Results** Performance of the VAE-$k$NN algorithm is exceptional (Area under the receiver-operating characteristics, AUC = 0.80) and remains stable across demographic and disease groups over the production period (AUC 0.74–0.82). We can detect issues in data feeds using our monitoring process to create immediate insights into future model performance.

**Conclusions** Our algorithm demonstrates exceptional performance at predicting risk of 30-day ED visits. We confirm that model outputs are equitable and stable over time using a proactive monitoring approach.

## Plain language summary

Patients with cancer often need to visit the hospital emergency department (ED), for example due to treatment side effects. Predicting these visits might help us to better manage the treatment of patients who are at risk. Here, we develop a computer-based tool to identify patients with cancer who are at risk of an unplanned ED visit within 30 days. We use health record data from over 28,000 patients who had visited a single cancer hospital in the US to create and test the model. The model performed well and was consistent across different demographic and disease groups. We monitor model behavior over time and show that it is stable. The approach we take to monitoring model performance may be a particularly useful contribution toward implementing similar predictive models in the clinic and checking that they are performing as intended.

[1] The Ronin Project, San Mateo, CA, USA. [2] Section of Patient-Centered Analytic, Division of Internal Medicine, The University of Texas MD Anderson Cancer Center, Houston, TX, USA. [3] The Lawrence J. Ellison Institute for Transformative Medicine, Los Angeles, CA, USA. [4] These authors contributed equally: Renee George, Benjamin Ellis. [5] These authors jointly supervised this work: Christine Swisher, Chris Sidey-Gibbons. ✉email: cgibbons@mdanderson.org

Modern cancer therapies are designed to eliminate disease and prevent recurrence. To this end, treatments including chemotherapy, immunotherapy, radiation, and surgery have proven to be largely effective with year-on-year reductions in mortality for most cancer types[1]. However, therapies are associated with negative effects that can range from mild to life threatening and may persist long after cancer treatment has been completed.

During therapy, patients with cancer often experience treatment-induced symptoms including fever, infection, pain, dehydration, neutropenia, and sepsis which may require urgent care in the emergency department (ED)[2]. Visits to the ED are common and between 41% and 64% of ED visits could be potentially avoidable as they are related to poorly-controlled disease or treatment-related symptoms such as fever, dehydration, and pain[2].

Better outpatient management of these symptoms can reduce the risk of ED visits and therefore unplanned ED visits could be reduced by accurate estimation and feedback of risk[3–5]. Correctly identifying patients at high risk of ED admission could facilitate timely therapies to reduce symptom burden and, if necessary, alterations to cancer therapy regimens. Increasing availability of patient care data via the electronic health record (EHR) as well as the proliferation of prediction tools have created an opportunity to guide individual risk assessment to provide timely management of treatment-related symptoms and to reduce the incidence of ED visits.

In recent years, studies have used a combination of EHR data and machine learning to predict ED admission within 7–180 days for cancer patients receiving diverse therapies[6–14]. Those studies have demonstrated moderate-to-good prediction performance measured using the area under the receiver-operator characteristic curve (AUC) of 0.62–0.75 on hold-out testing datasets. To our knowledge, no previous study in this area has evaluated or described a process for evaluating model performance once deployed in production, an important consideration for ML models.

Deploying ML models to perform live predictions must be done thoughtfully due to the complexity of the data which inform the model. Machine learning models are reliant on data and future changes in data quality, distribution, or definition may adversely affect model validity. Electronic health record data, which provides detailed patient histories and descriptions of patient interactions across health systems, may be especially sensitive to temporal variations in data distribution and quality as well as changing definitions across platforms. Because future data may differ, it is not sufficient to validate and 'freeze' machine learning models in time; evaluations of ML model performance must be made throughout the model lifespan[15].

Processes for real-time monitoring of system behavior combined with automated response is required for ongoing ML operations are well understood and implemented throughout the tech industry but are not currently well represented in the literature relating to AI-based clinical decision-making tools[16]. One challenge in assessing the ongoing validity of deployed AI-based clinical decision-making tools is that popular methods which compare predictions to ground truths (e.g., AUC) require the prediction timeframe (e.g., 30 days; 5 years) to have elapsed before ground truths can be known and model performance assessed.

The primary aim of this research was to develop a robust, reliable, and explainable model to predict 30-day ED admission while demonstrating a novel process for evaluating model behavior, including ongoing assessments of data quality and model performance, during a 3-month model deployment period in which the model was exposed to live data but outputs were not fed back to clinical staff.

We were able to produce a model with exceptional performance when predicting risk of 30-day ED risk on a hold-out testing data set. Our proactive 77-day in-production monitoring period confirms the stability of the algorithm with regards to its performance across demographic and disease groups.

## Methods

We recruited patients from the breast and genitourinary medical oncology at University of Texas MD Anderson Cancer Center, a specialist cancer center in the United States of America. Ethical approval was provided by the MD Anderson Cancer Center Institutional Review Board (IRB). Informed consent was waived by the IRB due to the planned retrospective analyses of routinely-collected clinical data. We used this data to train a ML model to predict whether a patient would visit the ED within 30 days.

**Featurization pipeline.** We developed a scalable featurization pipeline which could consistently transform EHR data into usable features. Features were derived primarily through the creation of semantic classes expressed through standard clinical terminologies. The classes were then applied to EHR data, largely Health Level 7 (HL7) standard Fast Healthcare Interoperability Resources (FHIR) resources, collected from Epic Systems. We calculated social determinants of health (SDoH) data using the definitions outlined in the United States Health and Human Services (HHS) Healthy People 2020 using both zip code & census tract data[17].

**Dataset (model training and testing).** We utilized data collected from patients attending GU and breast medical oncology departments at the study site. We aggregated data between April 1990 and April 2022 and created a featurization pipeline to continuously ingest and prepare future data for model predictions and evaluation. To ensure that our featurization process was scalable across institutions we exported data from the EHR in compliance with the 21st Century Cures Act and the United States Core Data for Interoperability (USCDI) standards[18,19]. Our dataset consisted of 184,138 observations from 28,369 patients. A new observation was defined every time a new lab value, vital sign, or staging result was detected in the EHR. The data were split into training and test samples using a 4:1 ratio. We ensured that no individual patient had observations in both training and testing sets.

We included a total of 222 features derived from 99 patient data elements, covering demographic, vital sign, laboratory, SDoH, comorbidity and cancer condition (cancer type and clinical and pathologic staging) data, that were chosen through exploratory analysis and feedback from clinical intelligence and informatics experts. We provided the description and distributions of the data elements included for model training in Supplementary Data 1 and 2. Categorical features were transformed using one-hot encoding and numeric features were either bucketized or min-max scaled. The featurization strategy automatically handled missingness by setting all relevant bucket columns or one-hot encoded columns to zero. Machine learning models were trained to predict whether a patient would visit the emergency department (ED) within 30 days following an observation. Further featurization details are presented in Supplementary Methods.

**Model training and testing.** We developed a variational autoencoder $K$-nearest neighbor (VAE-$k$NN) algorithm[20]. The latent

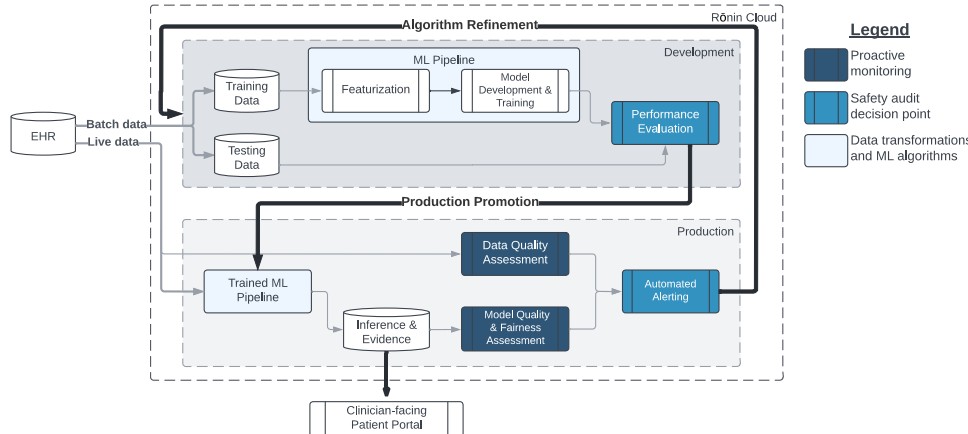

**Fig. 1 Overview of model development and model health monitoring strategy presented in this study.** Safety and quality were assured with Safety Audit decision points in both production and development (royal blue) and with proactive monitoring in production (navy blue). EHR: electronic health record, ML: machine learning.

feature vector from the VAE was fine-tuned with contrastive learning[21–23]. We chose this model due to both its high performance in predictive tasks as well as the intuitive way in which model outputs can be understood by describing the neighborhood that was used to make the prediction.

The VAE-$k$NN model was evaluated against a logistic regression with elastic net penalty (LR)[24], random forest (RF)[25], multinomial Naive Bayes (MNB)[26], $k$NN (without VAE encoding)[27], and gradient boosting machine (GBM) model[27]. We used Bayesian optimization with threefold cross-validation to tune the model hyperparameters using AUC as the performance criterion. Output probabilities were assessed the AUC and Brier score[28,29]. Further information relating to the VAE-$k$NN algorithm is given in Supplementary Methods.

**Model interpretability.** An essential component of ED risk assessments is the ability to explain predictions so that clinicians can intervene prior to unplanned ED admission. We used Shapley additive explanations (SHAP) to assign each feature an impact value for the entire training set as well as for individual predictions[17].

**Simulated model deployment.** Following successful model training and validation we put our validated production model into a simulated deployment lasting 77 days where it was exposed to live data, but model outputs were not fed back to clinical teams. The purpose of this simulated deployment was to observe model behavior in a real-world setting. We break down our approach into three principal elements to ensure reliable, reproducible, and safe delivery of ML predictions in practice. An overview of our strategy to monitor model health is shown in Fig. 1.

For a model to be deployed and remain in production, it must pass a series of internal checks. First, we expect that the overall performance of the model (measured by AUC) will not regress beyond a limit of 0.10 points. We also inspect the performance across several demographics (birth sex, race, ethnicity, and cancer type) to ensure that it follows the 4/5th rule (each demographic subgroup is within 20% of all others). Model performance, both overall and for different demographic or clinical groups, must remain within 0.10 AUC points of the originally validated model to remain in production. Models that violate any performance standards for 2 consecutive weeks will be investigated and may be retrained.

Central to our framework for continuous monitoring of model behavior and performance is the concept of lagging and leading indicators of model health (see Table 1). Lagging indicators, which include overall model prediction performance, are only available after the prediction time period (30 days in the current study) has elapsed and ground truth outcomes are known. Conversely, leading indicators are available prior to or immediately after a prediction is made.

Within this framework, a leading indicator is one that is immediately available (i.e., prior to the 30-day window to establish whether there had been an ED visit following a prediction). Leading indicators in our framework include data quality as well as the number and type of predictions made. Conversely, a lagging indicator can only be established once the prediction period has elapsed. The ability of leading indicators to signpost potential issues with model health before the ground truth is known may make them especially useful in ensuring safe and effective AI in practice.

Data drift, a leading indicator of model health, is measured using population stability index (PSI). We monitored PSI weekly during deployment as well as the volume and nature of predictions. After the true outcomes of patients were known we measured AUC overall performance as well as for the demographic subgroups on a weekly basis. Breaches in our data drift and performance metrics resulted in automated alerts notifying the Ronin team to investigate and resolve any issues that might affect model performance.

We evaluated bias during deployment using a combination of absolute and relative rules governing the model performance. If there was a 0.10 point reduction in AUC from baseline for the entire population or for any group divided by birth sex, race, ethnicity, or cancer type an alert was automatically generated. Similarly, the 4/5th rule is applied to ensure that the AUC in each demographic subgroup is within 80% of all others.

We conducted all analyses in the Python programming environment (Version 3.9) utilizing SciKit-Learn, HyperOpt; scikit-optimize; SHAP library, and TensorFlow/Keras. We reported our in accordance with the Transparent Reporting of a multivariable prediction model for Individual Prognosis Or Diagnosis the (TRIPOD) statement[30].

**Reporting summary.** Further information on research design is available in the Nature Portfolio Reporting Summary linked to this article.

**Table 1 Lagging and leading indicators of model health.**

| Type | Indicator | Description | Measurement metric | Function |
|---|---|---|---|---|
| *Leading indicators are available prior to when or immediately after predictions are made.* | | | | |
| Leading | Prediction volume | Number of predictions made | Volume | Relative changes can be monitored to indicate model is functioning as expected |
| | Data drift | Changes to availability, statistical properties, completeness, or transformations of data used to generate predictions | Population stability index, Descriptive statistics, missingness | Alterations in data distributions can signal changing phenomena or changes to labels or issues with featurization pipelines. |
| | Prediction drift | Changes to the distribution of predictions made by the model | Descriptive statistics | Proportional changes to model predictions (e.g., average predicted risk) can signal changes within data |
| *Lagging indicators are available only after ground truth outcomes are known about the predictions (more than 30 days).* | | | | |
| Lagging | Label drift | Proportion of ground truth outcomes associated with predictions | Descriptive statistics | Understand the number of true events occurring while model is in production |
| | Discrimination | Changes to model performance evaluated | AUC | Evaluate model discriminatory performance in production |
| | Algorithmic bias | Evidence that algorithm is functioning equivalently between demographic and clinical groups | AUC by group and difference between best and worst performing group | Ensure unbiased predictions across demographic and clinical groups |
| | Calibration | Changes to calibration | Brier Score | Evaluate changes to calibration in production |

AUC Area under the receiver operating characteristics curve.

## Results

**Description of our sample.** We assembled 184,138 observations that were collected from the Epic EHR. These records represented details of medical history and care received for 28,369 unique patients, a total of 16,000 (8.75%) observations were made within 30 days of an ED visit.

Demographic information is presented in Table 2. Most observations were taken from patients with a diagnosis of GU cancer (74,215; 50%) with 51,507, (35%) observations taken from patients with a diagnosis of breast cancer. The average age of participants in our sample was 61 (standard deviation = 13) and 53% were male. Within our sample, 28,307 observations were made from patients who had attended appointments at either the GU or breast departments but did not have a recorded diagnosis of breast or GU cancer.

**Featurization.** Details of the features developed using our scalable featurization pipeline including details of bucketization are presented in Supplementary Methods and Supplementary Data 1, 2, and 3.

**Model development and training.** All the models we tested performed well on the testing dataset (AUC range 0.76–0.80; see Supplementary Table 1) and were well calibrated (Brier Score range 0.07–0.13). Our preferred VAE-$k$NN algorithm was, alongside the gradient-boosted machines, the strongest performing algorithm (see Fig. 2a and b). The VAE-$k$NN performed equitably across all sensitive groups (Fig. 2c).

**Model interpretability.** Interpretability visualizations are provided in Fig. 3. Figure 3a displays the mean SHAP value for the top 20 features used by the VAE-$k$NN. We present two separate use cases relating to an ED visit within 30 days; Fig. 3b, c and no ED visit within 30 days; Fig. 3d, e. Figure 3b, d present a visual confirmation of the proportion of patients in the VAE-kNN prediction neighborhood who experienced an ED visit. Figure 3c, e present SHAP values for the individual patient the prediction was made for.

**Data health monitoring.** Our weekly assessments of model health throughout the 77-day production assessment period indicated acceptable PSI (Fig. 4b) for most weeks. We were able to detect a 2-week period in which prediction volume was far lower due to issues with the application processing interface (API) connection to the EHR. This issue was resultingly resolved using the system shown in Fig. 1.

**Performance and fairness monitoring.** We monitored lagging indicators of model health as soon as the ground truth information was available within the EHR to determine if the patient had gone to the ED or not. Table 3 displays average performance scores over the deployment period. Figure 4d demonstrates that there were fewer ED visits for patients in the production period than were represented in the training datasets. The reduced number of visits to the ED may have been a driving factor in our discrimination performance dropping slightly, on average, across the production assessment period (Fig. 4e). The calibration of our model was maintained, compared to testing, across the production period (Fig. 4f).

## Discussion

In this paper we demonstrate cutting edge performance in the task of predicting 30-day ED use for cancer patients while providing a novel method for communicating model outputs to clinicians. We introduce the concept of leading and lagging indicators and demonstrate how model health can be constantly

**Table 2 Descriptive statistics for all patients involved in our study.**

|  |  | Train | | Test | | Production | |
|---|---|---|---|---|---|---|---|
|  |  | *N* | % | *N* | % | *N* | % |
| Metadata | Total observations | 147,023 |  | 37,115 |  | 14,465 |  |
|  | Individual patients | 22,701 |  | 5668 |  | 7174 |  |
|  | Observations within 30 days of ED visit | 12,719 | 8.65 | 3281 | 8.84 | 1008 | 6.97 |
| Birth sex | Unknown | 199 | 0.14 | 13 | 0.04 | 14 | 0.1 |
|  | Male | 77,919 | 53 | 18,611 | 50.14 | 6933 | 47.93 |
|  | Female | 68,711 | 46.73 | 18,450 | 49.71 | 7516 | 51.96 |
| Race | White/Caucasian | 110,697 | 75.29 | 27,251 | 73.42 | 11,254 | 77.8 |
|  | Black/African American | 16,306 | 11.09 | 4394 | 11.84 | 1422 | 9.83 |
|  | Asian | 6498 | 4.42 | 1863 | 5.02 | 657 | 4.54 |
|  | Other | 13,522 | 9.2 | 3607 | 9.72 | 1132 | 7.83 |
| Ethnicity | Unknown | 3253 | 2.21 | 638 | 1.72 | 325 | 2.25 |
|  | Not Hispanic/Latino | 122,234 | 83.14 | 30,442 | 82.02 | 12,236 | 84.59 |
|  | Hispanic/Latino | 20,850 | 14.18 | 5851 | 15.76 | 1902 | 13.15 |
| Cancer Dept | GU | 74,215 | 50.48 | 17,744 | 47.81 | 5837 | 40.3 |
|  | Breast | 51,507 | 35.03 | 14,027 | 37.79 | 5626 | 38.89 |
|  | Other | 22,693 | 15.43 | 5614 | 15.13 | 3101 | 21.44 |

*ED* emergency department, *GU* Genitourinary.

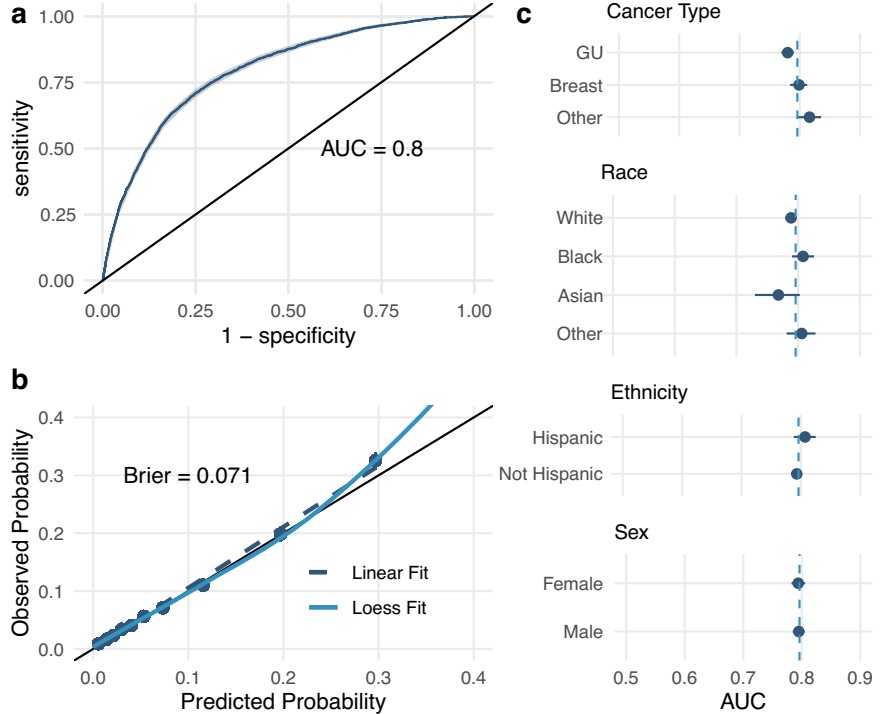

**Fig. 2 Model performance in held-out test data set. a** Overall Discrimination; **b** Overall Calibration; **c** Discrimination across sensitive groups. AUC: area under the receiver operating characteristic curve, GU: Genitourinary. Source data is in Supplementary Data 4. Error bars represent 95% confidence interval. N for analyses shown in Table 2.

monitored before the 30-day prediction period has elapsed and ground truth data becomes available. As far as we are aware, this is the first example of a model which has been developed, deployed, and subjected to an extensive period of comprehensive data quality, model behavior, and bias monitoring.

We demonstrate that our algorithm performs at a greater or equal level to other published oncology prediction models developed using the EHR to predict unplanned ED use in either general oncology populations or specific cancer types or treatments[6–14]. Uniquely, we are able to demonstrate the consistency of this performance on live data during our production implementation. In addition, we introduce the concept of leading indicators of model health as tools to generate rapid diagnostic to support longer prediction timeframe and safety of the delivered predictions[12].

At the core of our work was the desire to make an algorithm that demonstrated good performance during development and validation, but which also was able to maintain that performance during production where it was monitored in live deployment at an academic medical center in the United States. Consistent with our expectations, overall model performance did not drop-off during production.

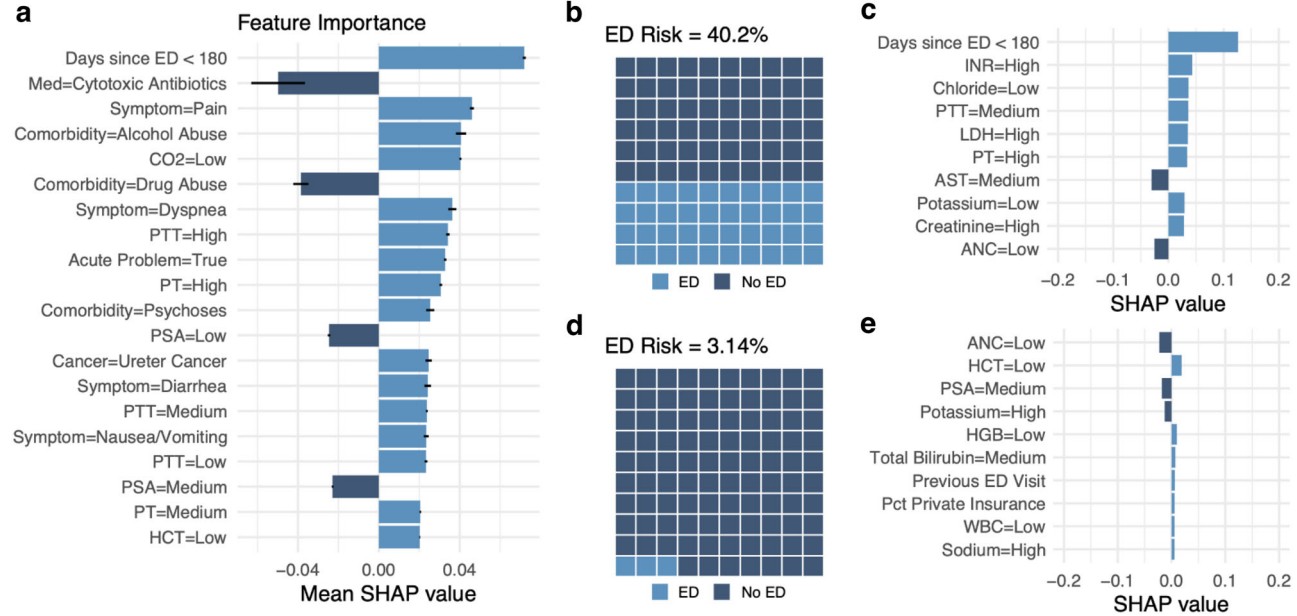

**Fig. 3 Model explainability/interpretability. a** Overall feature/value importance. **b, c** High risk example patient. **d, e** Low risk example patient. **b/d** Predicted risk displayed as an easy-to-understand pictogram. **c/e** Top 10 feature/value contributions for each example patient. ED: emergency department, SHAP: Shapely addictive explanations. Med: Medicine, CO2: Carbon dioxide, PTT: Partial thrombastin time, PT: Prothrombin time, PSA: Prostate specific antigen, HCT: Hematocrit, INR: International normalized ratio, LDH: Low-density lipoprotein, AST: Aspartate aminotransferase, ANC: Neutrophil count, WBC: White blood count, HGB: Hemoglobin, Pct: Percentage. Source data is in Supplementary Data 4. Error bars represent 95% confidence interval.

Maintaining model performance is dependent on maintaining data and code quality. These three dependent concepts are a requirement for delivering high quality models. During the weeks of May 16th and May 23rd, 2022 we noticed that the number of predictions dropped sharply, as did our average predicted risk. Conversely, our data drift monitor spiked. We note that this issue would likely not have been noticed if we were monitoring for model performance alone. An investigation found that data quality had dropped due to outages in EHR data feeds that the presented ED risk model relies upon. Our model was receiving incomplete data for the predictions performed during this time span. Using the data health monitors in place and collaboration between the data teams at both institutions, the issues were quickly resolved, which is reflected in our data health metrics returning to their expected ranges.

Our process for monitoring data quality proved useful in this study where our AI system was put into production, but predictions were not fed back to providers. The importance of model behavior and data quality monitoring will increase when the predictions are fed back to clinicians as part of a decision support system. In such a system, we expect that the number of high-risk patients that are admitted to the ED will decrease as effective clinical interventions are deployed as a result of the model predictions. The expected decrease in true positive rate will make it impossible to monitor model performance by relying only on lagging indicators of prediction quality such as AUC, as other studies have done[14].

Further research is required to assess the impact of the model upon clinical decision making and subsequent improvements in patient outcomes resulting from the model. New techniques to interpret leading indicators of model health will allow greater insight into how changes in data distributions and model behavior around the time of prediction might impact model performance without having to rely on lagging indicators which may not be useful in the case of an in-production model. In

addition, another key component of an effective ML model is the communication of its outputs to the target audience. In this work, we used a *k*NN algorithm which has unique properties to offer ways of model output communications to clinicians and patients. To maintain this paper in a manageable length, we focused this paper on the performance of the models and our ability to monitor their behavior over time. Another paper describing how to leverage the properties to design the model output feedback system with the involvement of end-users, including clinicians, nurses, and oncologists, is under preparation.

We acknowledge some limitations in our work; we were able to produce a model which was found to be suitable for use at two departments within a single large academic cancer center in the United States. We attempted to deploy informatics processes which could be standardized across sites using similar data sources in the United States, but were unable to assess whether our featurization, development, and validation pipelines could create acceptable models at other institutions or for other disease sites. In addition, studies have shown that other forms of data which are not necessarily linked to the EHR, such as patient-reported outcomes and radiation parameters, may provide a strong signal and improve the performance of prediction models and their inclusion may well have improved the performance of our model for this task.

In concusion, we present a strategy for ensuring that ML models can be develop, deployed, and proactively monitored to ensure consistent unbiased performance during the model lifetime, as far as we are aware, our approach is novel in the healthcare setting. Our model that displayed state-of-the-art performance on the prediction task and maintained the equivalent performance across demographic groups when exposed to live data over a period of 3 months. Future work should be conducted to estimate the performance of this pipeline in other settings as well as to create new strategies to identify threats to

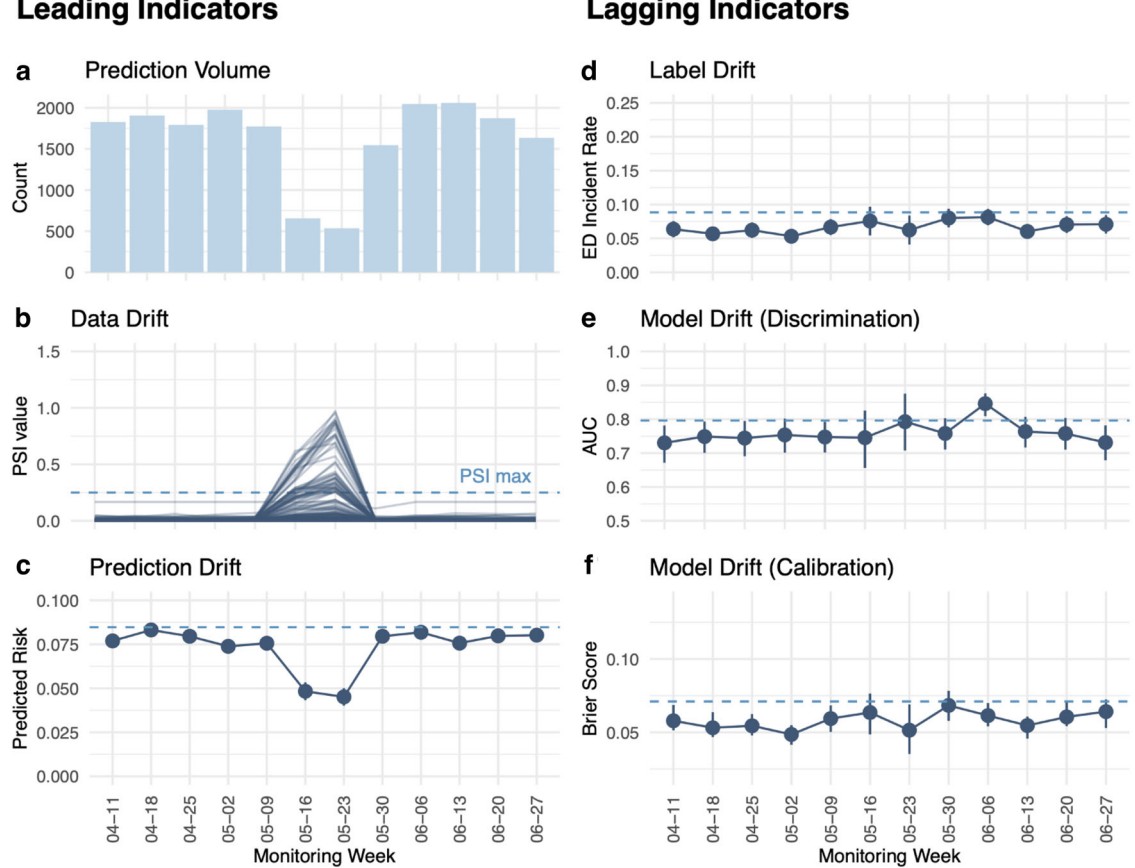

**Fig. 4 Leading and lagging monitoring metrics in production.** Weekly metrics used to assess data health and model performance. Leading metrics: **a** Volume of predictions. **b** PSI value for all features. **c** Average predicted risk. Lagging metrics: **d** Percent of actual ED visits. **e**, **f** AUC and Brier scores. The drop in volume and spike in PSI was due to an EHR data feed outage and is described in the *Discussion* section. PSI: population stability index. Source data is in Supplementary Data 4. Error bars represent 95% confidence interval, N for analysis shown in (**b**–**f**) are given in (**a**).

**Table 3 Performance of the VAE-*k*NN algorithm on the testing and production datasets.**

| | | AUC [95% CI] | | Delta |
|---|---|---|---|---|
| | | **Test** | **Production[a]** | |
| | Overall | 0.80 [0.79, 0.80] | 0.76 [0.74, 0.77] | −0.04 |
| Birth sex | Male | 0.79 [0.78, 0.81] | 0.78 [0.75, 0.80] | −0.02 |
| | Female | 0.79 [0.78, 0.81] | 0.74 [0.72, 0.76] | −0.05 |
| Race | White/Caucasian | 0.79 [0.78, 0.80] | 0.75 [0.74, 0.77] | −0.03 |
| | Black/African American | 0.81 [0.79, 0.83] | 0.74 [0.70, 0.79] | −0.06 |
| | Asian | 0.77 [0.73, 0.80] | 0.77 [0.71, 0.84] | 0.01 |
| | Other | 0.80 [0.78, 0.83] | 0.77 [0.71, 0.82] | −0.03 |
| Ethnicity | Unknown | 0.77 [0.74, 0.86] | 0.82 [0.74, 0.90] | 0.05 |
| | Not Hispanic/Latino | 0.79 [0.78, 0.80] | 0.76 [0.74, 0.78] | −0.03 |
| | Hispanic/Latino | 0.81 [0.79, 0.82] | 0.71 [0.66, 0.76] | −0.10 |
| Cancer Dept | GU | 0.78 [0.77, 0.79] | 0.76 [0.74, 0.79] | −0.02 |
| | Breast | 0.80 [0.78, 0.81] | 0.73 [0.71, 0.76] | −0.07 |

*AUC* area under the receiver operating characteristic curve, *GU* Genitourinary.
[a]Production values are averaged from weekly observations across the 67-day production period.

model performance via leading indicators of model health and behavior.

### Data availability
All data relating to the study are presented in the paper and supplementary materials. Patient data used to create, validate, and monitor our models are not publicly available due to ethical concerns. It may be possible to make data available as part of a future academic collaboration requiring separate institutional data collaboration agreements and additional IRB approval. Please contact Dr. C Gibbons to discuss. Source data for the figures are available as Supplementary Data 4.

### Code availability
Analysis code can be accessed from the following repository:[31] https://zenodo.org/record/7888547 (https://doi.org/10.5281/zenodo.7888547)

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

## Acknowledgements

Hao Sun and Kurt Weber for generating value sets required for featurization. Natalie Jeha and Michelle Abramowski for contributing to feature selection and output vetting. Max Kaufmann and Neehar Mukne for contributing NLP models to identify ED visits from EHR notes.

## Author contributions

R.D.G. and C.L.S. conceptualized and managed the study. A.G. and A.W. developed featurization software. S.W., M.A., and K.B. provided clinical expertize and developed value sets for medical codes. R.D.G., B.H.E., A.G., and A.W. performed model training and validation, SHAP computations and analysis, and developed monitoring software. R.D.G., B.H.E., and L.K. performed analysis of results. C.L.S., R.D.G., and B.H.E. co-authored the paper and generated graphics and figures. C.S.G. and S.C.L. provided clinical expertize and co-authored the paper. C.S.G. and S.C.L. performed literature review.

## Competing interests

The authors declare the following competing interests: R.G., B.E., A.W., A.G., S.W., M.A., K.B., L.K., and C.S. all have stock options in Project Ronin, who funded this research. C.S.G. has received research funding, including salary support, for Project Ronin. Other authors report no competing interests.
