## [Peer Review File · Communications Medicine]

Reviewers' comments:

Reviewer #1 (Remarks to the Author):

Thank you for the opportunity to review this manuscript.

The manuscript describes a model for predicting whether individuals in a select population of oncology patients will present to the emergency department during their therapy, through the use of a variational autoencoder k nearest neighbors algorithm. The paper has several strengths. The authors' approach is sound and well-delineated, with detailed reporting of the feature variables and model design. The model's performance demonstrates a small improvement relative to the studies cited by the authors, and has the benefit of balanced performance across demographic groups. The authors provide several comparisons with alternate methodologies, and appropriately note a rough equivalence of these approaches. Finally, the use of regular model updates and longitudinal measurement of performance is innovative and should be more frequently reported in the literature.

However, there are some issues with the manuscript. Specific comments follow:

Lines 29-31 - "Visits to the ED are common and between 41% and 64% of ED visits could be potentially avoidable as they are related to poorly-controlled disease or treatment-related symptoms such as fever, dehydration, and pain." You have identified that a significant proportion of visits to the ED in this population are avoidable; however, your model appears to predict only whether patients will present to the emergency department, not whether the visit itself is avoidable.

Lines 39-40 - "In recent years, studies have used a combination of EHR data and machine learning to predict ED admission within seven to 180 days for cancer patients receiving diverse therapies" - the language used here is vague - are you describing presentation to the emergency department for symptom management, or admission from the emergency department to the hospital?

Lines 115-117 - "In this paper we demonstrate cutting edge performance in the task of predicting 30-day ED use for cancer patients while providing a novel method for communicating model outputs to clinicians" - Is your claim that the use of Shapely Additive Explanations is a novel methodology? While the use of explainable ML techniques in clinical deployment is to be encouraged, there are several existing clinical studies utilizing this technique alone (viz. Chowdhury SU, et al. Shapley-Additive-Explanations-Based Factor Analysis for Dengue Severity Prediction using Machine Learning. Journal of Imaging. 2022 Aug 26;8(9):229., Nordin N, et al. An explainable predictive model for suicide attempt risk using an ensemble learning and Shapley Additive Explanations (SHAP) approach. Asian journal of

psychiatry. 2023 Jan 1;79:103316., Nohara Y, et al. Explanation of machine learning models using shapley additive explanation and application for real data in hospital. Computer Methods and Programs in Biomedicine. 2022 Feb 1;214:106584.)

Reviewer #2 (Remarks to the Author):

This paper describes the development, validation and monitoring of a predictive model for ED admission for breast and genitourinary cancer patients. Methods for monitoring deployed models is important. However, the paper is lacking in methodological details which are important in order to appropriately understand and evaluate the model development and implementation approach. There is also a lack of attention to CDS implementation methods, such as design and feedback with end-users, which is critical in this type of work. The authors likely need to clarify the scope of the paper, but regardless the lack of end-user involvement/input for the development and implementation of a model being evaluated in the clinical setting concerning. Please see comments below:

1. Who are the model end-users?
2. Please restructure so that methods are presented before results, as is standard
3. EHR data is complex and entered by multiple different types of clinicians. It is not clear what the scope of the EHR data being used. There is very limited description of the EHR data used in this study. Please described the data set in detail - type of site(s), patient population, setting (inpatient, outpatient, ED, all?), scope of clinical data elements used (e.g., include flowsheet data, medication administration record data, notes?), data documented by which clinical roles? (e.g., physicians, nurses, physical therapists, social workers? Etc).
4. Please include a descriptive name and definition for each feature in eTable 1. Providing the abbreviated name from your research database is not useful to a reader and does not allow for accurate interpretation.
5. The methods state that when new data is entered into the EHR it is then included in the model. How did the investigators account for healthcare processes driving the temporal nature of patient assessment and data collection in the EHR? The presence of a clinical data point may be more likely for patients experiencing poor outcomes/increased symptoms. In other words, were the data used in your study only collected from routine standard, scheduled screenings or was some data only collected when patients were sick? There is a lack of discussion around this notion. How is this handled/used/leveraged in the model. Please discuss implications for model interpretation and use.

6. Please provide evidence to back up statements at bottom of page 6. There is limited evidence of impact from AI models in the clinical setting to date and these claims appear unfounded.

7. There are many scientific steps required for implementation and evaluation of impact of a model on decision making, including methods for user-centered design for CDS. The role of CDS design is not mentioned in the manuscript. Please address and explain.

8. There appears to be no end-user design or feedback. This is a major limitation given the focus on Shapley for model explanation. How were end-users involved in this study? Who are the end-users? Physicians, nurses, care managers, home health?

9. Page 7 lines 158-167 - first time the settings were described. Please describe earlier and in more detail.

10. page 11 line 253 Please define data quality and how it was assessed.

11. You appear to evaluate bias within groups pre to post, but not between groups. EHR data is known to be highly biased and when used in models without bias mitigation approaches leads to biased model outputs. How are these model biases evaluated and handled in order to produce an equitable model output?

12. Please comment on the approach of using 1 model for 2 types of cancer? Is goal of this model to make it generalizable to any type of cancer? If not what is rationale for combining these 2 types of cancer?

February 21, 2023

Thank you for the opportunity to submit a revision of our manuscript entitled “Ensuring fair, safe, and interpretable artificial intelligence-based prediction tools in a real-world oncological setting [COMMSMED-22-0449A]” to Communications Medicine. We appreciate the time and effort you have dedicated to providing valuable feedback. We incorporated changes to reflect all of the suggestions provided, and below is a point-by-point response to the comments and concerns. Changes to the manuscript are presented in emboldened and underlined text.

Reviewers’ comments:

Reviewer #1 (Remarks to the Author):

Thank you for the opportunity to review this manuscript.

The manuscript describes a model for predicting whether individuals in a select population of oncology patients will present to the emergency department during their therapy, through the use of a variational autoencoder k nearest neighbors algorithm. The paper has several strengths. The authors’ approach is sound and well-delineated, with detailed reporting of the feature variables and model design. The model’s performance demonstrates a small improvement relative to the studies cited by the authors and has the benefit of balanced performance across demographic groups. The authors provide several comparisons with alternate methodologies, and appropriately note a rough equivalence of these approaches. Finally, the use of regular model updates and longitudinal measurement of performance is innovative and should be more frequently reported in the literature.

However, there are some issues with the manuscript. Specific comments follow:

1. Lines 29-31 - “Visits to the ED are common and between 41% and 64% of ED visits could be potentially avoidable as they are related to poorly-controlled disease or treatment-related symptoms such as fever, dehydration, and pain.” You have identified that a significant proportion of visits to the ED in this population are avoidable; however, your model appears to predict only whether patients will present to the emergency department, not whether the visit itself is avoidable.

Response: The Reviewer is correct. We sought to predict the risk of an ED visit, not whether that visit might be avoidable. To train a model to predict avoidable visits only would require us to manually re-code each of the 16000 observations made within 30-days of an ED visit within our dataset and specify whether we thought they may be avoidable. As well as being unscalable, this approach is fraught with risk of introducing additional biases. Our intention for the algorithm is that the risk information is fed back to clinical teams alongside the

rationale for the decision (e.g., lab values at a certain level) to inform changes to clinical management which may reduce the risk of ED visits.

2. Lines 39-40 - "In recent years, studies have used a combination of EHR data and machine learning to predict ED admission within seven to 180 days for cancer patients receiving diverse therapies" - the language used here is vague - are you describing presentation to the emergency department for symptom management, or admission from the emergency department to the hospital?

Response: We thank the Reviewer for this observation. They are correct insofar the language being vague – the included papers variously describe both admission both *to* and *from* the ED. We have now amended the text to read more clearly "... machine learning to predict admission to and from the ED within seven to 180 days...". We appreciate that while the wording may be subtly different the clinical reality between these two groups of patients may markedly different. Nonetheless, we would like to keep all references the same because prediction performance did not differ substantially between the two types of admission.

3. Lines 115-117 - "In this paper we demonstrate cutting edge performance in the task of predicting 30-day ED use for cancer patients while providing a novel method for communicating model outputs to clinicians" - Is your claim that the use of Shapely Additive Explanations is a novel methodology? While the use of explainable ML techniques in clinical deployment is to be encouraged, there are several existing clinical studies utilizing this technique alone (viz. Chowdhury SU, et al. Shapley-Additive-Explanations-Based Factor Analysis for Dengue Severity Prediction using Machine Learning. Journal of Imaging. 2022 Aug 26;8(9):229., Nordin N, et al. An explainable predictive model for suicide attempt risk using an ensemble learning and Shapley Additive Explanations (SHAP) approach. Asian journal of psychiatry. 2023 Jan 1;79:103316., Nohara Y, et al. Explanation of machine learning models using shapley additive explanation and application for real data in hospital. Computer Methods and Programs in Biomedicine. 2022 Feb 1;214:106584.)

Response: The Reviewer is absolutely correct. The SHAP technique is not novel. However, the feedback technique we were referring to was specific to the *k*NN algorithm we used – a process which would allow us to communicate uncertainty by describing the neighbourhood used to make the prediction. Given the complexity of this feedback methodology we eventually decided not to focus on it within the current paper but had incorrectly kept the claim of novelty. As such, we have edited the sentence as shown below:

"In this paper we demonstrate cutting edge performance in the task of predicting 30-day ED use for cancer patients."

Reviewer #2 (Remarks to the Author):

This paper describes the development, validation and monitoring of a predictive model for ED admission for breast and genitourinary cancer patients. Methods for monitoring deployed

models is important. However, the paper is lacking in methodological details which are important in order to appropriately understand and evaluate the model development and implementation approach. There is also a lack of attention to CDS implementation methods, such as design and feedback with end-users, which is critical in this type of work. The authors likely need to clarify the scope of the paper, but regardless the lack of end-user involvement/input for the development and implementation of a model being evaluated in the clinical setting concerning. Please see comments below:

1. Who are the model end-users?

Response: The intended model end users are medical oncologists, oncology nurse practitioners, oncology physician assistants, oncology nurse navigators and oncology triage nurses.

2. Please restructure so that methods are presented before results, as is standard

Response: This formatting convention is common to many journals within the Nature portfolio. We agree with the Reviewer that it is not standard but request to leave it in this format to remain in line with the guidance provided for Authors.

3. EHR data is complex and entered by multiple different types of clinicians. It is not clear what the scope of the EHR data being used. There is very limited description of the EHR data used in this study. Please described the data set in detail - type of site(s), patient population, setting (inpatient, outpatient, ED, all?), scope of clinical data elements used (e.g., include flowsheet data, medication administration record data, notes?), data documented by which clinical roles? (e.g., physicians, nurses, physical therapists, social workers? Etc).

Response: The EHR data used was extracted from a large oncology centre in the American Southwest. Inclusion was intentionally non-restrictive and included patient records, within the bounds of a mutual data use agreement, for any patient treated by two specific medical oncology services: Breast or GU. While the specific organization of the oncology services represents an institutional construct, the diagnosis and populations are reproducible in hospitals across the US by including those with cancer originating in the genitourinary organs or breasts who participating in active medical management.

The EHR data was obtained as FHIR resources from the EHR FHIR exchange APIs. The specific FHIR model was STU3. This approach was intentional as it represents the best chance for interoperability and generalization in post 21st Century Cures Act world as well mitigating the inherent process at any healthcare institute by consuming data transformed into federally endorsed standard model.

For each patient, a longitudinal extract of FHIR resources included Observations (Labs, Vitals), Medications (Medications, referenced Medication Entities, Medication Administrations), Conditions (Problem List Items and Encounter Diagnosis), Encounters, Documents and Patients (demographics for consideration of Social Determinants) was obtained. Non-FHIR resources included EHR Oncology Module Treatment plan extracts (cycle day, cycle number, treatment line) and Appointments.

4. Please include a descriptive name and definition for each feature in eTable 1. Providing the abbreviated name from your research database is not useful to a reader and does not allow for accurate interpretation.

Response: We appreciate the suggestion. We have included a new table (eTable 2) describing the descriptive name and definition for each feature in the manuscript.

5. The methods state that when new data is entered into the EHR it is then included in the model. How did the investigators account for healthcare processes driving the temporal nature of patient assessment and data collection in the EHR? The presence of a clinical data point may be more likely for patients experiencing poor outcomes/increased symptoms. In other words, were the data used in your study only collected from routine standard, scheduled screenings or was some data only collected when patients were sick? There is a lack of discussion around this notion. How is this handled/used/leveraged in the model. Please discuss implications for model interpretation and use.

Response: We agree with the Reviewer that the presence of absence of data may be predictive and that, the best performing models may certainly have the most complete data. However, a key goal of our paper was to demonstrate a tool that would integrate to current clinical data acquisition systems and therefore did not require additional data collection.

We developed our method to reduce potential fluctuations in model behaviour as data fields became sparse or saturated at each time point. We opted to carry data forward over a reasonable time span (e.g., 7 days for symptoms and labs, 6 months for comorbidities, cancer type, and medication) to avoid issues of sparsity in observations. We have now listed this in our Methods section which now reads:

“The featurization strategy automatically handled missingness by setting all relevant bucket columns or one-hot encoded columns to zero. Where possible, we used a fill-forward methodology to create data for observations. For symptoms and lab result we limited the fill-forward period to 7 days, and for comorbidities, cancer type, and medication we set the fill-forward limit to 6 months.”

We have added a sentence to our discussion regarding the pragmatic use of clinical data.

“We note that the pragmatic nature of the data which was collected for other purposes meant that it was incomplete and may contain a number of biases, such as availability bias, which ultimately reduce performance.”

6. Please provide evidence to back up statements at bottom of page 6. There is limited evidence of impact from AI models in the clinical setting to date and these claims appear unfounded.

Response: We agree. We intended to talk hypothetically to explain how if clinical interventions were successful, then the outcomes would improve, changing the ground truths which would make it appear as the model was underperforming if we were to compare

predictions to ground truth outcomes. We did not mean to suggest that clinical interventions were likely to be successful and have changed the text to avoid that interpretation thus:

“If such a system were to work effectively as expected, we would witness a reduction in the number of patients who are flagged as high-risk and then are admitted to the ED as an effective clinical intervention would improve patient health and avoid the ED visit. The expected decrease in true positive rate will make it impossible to monitor model performance by relying only on lagging indicators of prediction quality such as AUC, as other studies have done. ¹⁴”

7. There are many scientific steps required for implementation and evaluation of impact of a model on decision making, including methods for user-centered design for CDS. The role of CDS design is not mentioned in the manuscript. Please address and explain.

Response: Again, we agree that the process towards proper and effective implementation requires a series of steps beyond the demonstration of validity – such as the way in data and interfaces are displayed and integrated within the clinical workflow. Such work was beyond the remit of the current paper but we have added to our Discussion section to clarify some additional work that can be done.

“To facilitate effective intervention of AI-based CDSS, further research is required to understand how risk predictions can be best fed-back to stakeholders and integrated into clinical workflows. Additional research to demonstrate the positive impact of the model upon clinical decision making and subsequent improvements in patient outcomes resulting from the model are required before widespread adoption of AI-based CDSS can be a reality.”

8. There appears to be no end-user design or feedback. This is a major limitation given the focus on Shapley for model explanation. How were end-users involved in this study? Who are the end-users? Physicians, nurses, care managers, home health?

Response: We did include users in the design of the feedback system details but wanted to focus this paper on the performance of the models and our ability to monitor their behaviour over time. For the feedback design we recruited 40 clinicians, 30 nurses and 10 oncologists. A further paper will describe the results of this investigation.

9. Page 7 lines 158-167 - first time the settings were described. Please describe earlier and in more detail.

Response: We appreciate the suggestion and have revised the first two sentences of the result section to “We assembled 184,138 observations that were collected from the Epic EHR of a single large academic cancer in the United States. These records represented details of medical history and care received for 28,369 unique patients from the breast and genitourinary medical oncology, a total of 16,000 (8.75%) observations were made within 30 days of an ED visit.” to describe our settings earlier.

10. page 11 line 253 Please define data quality and how it was assessed.

Response: We acknowledge the ambiguity of the term “data quality”, especially in this context. Upon reflection, we believe that the term “consistency” is more appropriate than “quality” and have changed these terms throughout the manuscript.

11. You appear to evaluate bias within groups pre to post, but not between groups. EHR data is known to be highly biased and when used in models without bias mitigation approaches leads to biased model outputs. How are these model biases evaluated and handled in order to produce an equitable model output?

Response: We agree that bias mitigation and evaluation are important to produce equitable model outputs. We have added details of our between-group bias evaluation in Table 2. shown below. All results fall within the 4/5th rule.

	AUC [95% CI]		AUC Min:Max Ratio		Delta
	Test	Production*	Test	Production*	
Overall	0.80 [0.79, 0.80]	0.76 [0.74, 0.77]			-0.04
Birth sex					
Male	0.79 [0.78, 0.81]	0.78 [0.75, 0.80]	0.99	0.95	-0.02
Female	0.79 [0.78, 0.81]	0.74 [0.72, 0.76]			-0.05
Race					
White/Caucasian	0.79 [0.78, 0.80]	0.75 [0.74, 0.77]			-0.03
Black/African American	0.81 [0.79, 0.83]	0.74 [0.70, 0.79]	0.95	0.96	-0.06
Asian	0.77 [0.73, 0.80]	0.77 [0.71, 0.84]			0.01
Other	0.80 [0.78, 0.83]	0.77 [0.71, 0.82]			-0.03
Ethnicity					
Unknown Unreported	0.77 [0.74, 0.86]	0.82 [0.74, 0.90]			0.05
Not Hispanic/Latino	0.79 [0.78, 0.80]	0.76 [0.74, 0.78]	0.95	0.87	-0.03
Hispanic/Latino	0.81 [0.79, 0.82]	0.71 [0.66, 0.76]			-0.10
Cancer Dept					
GU	0.78 [0.77, 0.79]	0.76 [0.74, 0.79]	0.98	0.96	-0.02
Breast	0.80 [0.78, 0.81]	0.73 [0.71, 0.76]			-0.07

12. Please comment on the approach of using 1 model for 2 types of cancer? Is goal of this model to make it generalizable to any type of cancer? If not what is rationale for combining these 2 types of cancer?

Response: We had the opportunity to develop a model with large amounts of EHR data from two Departments. This meant that we developed our model to predict ED issues related to a number of different cancers that were treated at the Genitourinary and Breast Departments. The goal is a generalizable model that is workable across cancer types.

Different models are not needed for each cancer type because types are treated as features in our model. We have added this to the Discussion section:

“We attempted to deploy informatics processes which could be standardized across sites using similar data sources in the United States, but were unable to assess whether our featurization, development, and validation pipelines could create acceptable models at other institutions or for other disease sites. Our goal is to expand our pipeline to add additional disease sites as the deployment of the model matures at our institution”

REVIEWERS' COMMENTS:

Reviewer #1 (Remarks to the Author):

Thank you for addressing the questions raised in the review.

Reviewer #2 (Remarks to the Author):

Authors responses are appropriate and adequate to address reviewers concerns. However, two responses are still not incorporated into the manuscript for all readers.

1. description of the data used, including the specific EHR data types. This information is essential for reproducibility.
2. clarification that end-users were involved in the feedback system design, but that those data will be presented in a separate publication.

Reviewer 2:

Authors responses are appropriate and adequate to address reviewers concerns. However, two responses are still not incorporated into the manuscript for all readers.

1. description of the data used, including the specific EHR data types. This information is essential for reproducibility.

Response: We have now included this information both in the Method section and Supplementary Data 1 & 2.

2. Clarification that end-users were involved in the feedback system design, but that those data will be presented in a separate publication.

Response: We have included this information in the Discussion section of the manuscript which reads: "In this work, we used a kNN algorithm which has

unique properties to offer ways of model output communications to clinicians and patients. To maintain this paper in a manageable length, we focused this paper on the performance of the models and our ability to monitor their behavior over time. Another paper describing how to leverage the properties to design the model output feedback system with the involvement of end-users, including clinicians, nurses, and oncologists, is under preparation.“